# Benefits of Coherent Large Beamwidth Processing of Radio-Echo Sounding Data

Anton Heister[1] and Rolf Scheiber[1]

[1]German Aerospace Center (DLR), Microwaves and Radar Institute, Wessling, Germany

**Correspondence:** Anton Heister (anton.heister@dlr.de)

**Abstract.** Coherent processing of radio echo sounding data of polar ice sheets is known to provide indication of bedrock properties and detection of internal layers. We investigate the benefits of coherent processing of a large azimuth beamwidth to retrieve and characterize the orientation and angular backscattering properties of the surface and subsurface features. MCRDS data acquired over two distinct test areas in Greenland are used to demonstrate the specular backscattering properties of the ice surface and of the internal layers, as well as the much wider angular response of the bedrock. The coupling of internal layers' orientation with the bed topography is shown to increase with depth. Spectral filtering can be used to increase the SNR of the internal layers and for mitigating the surface multiple. The variance of the bed backscattering can be used to characterize the bed return specularity. The use of the SAR focused RES data ensures the correct azimuth positioning of the internal layers for the subsequent slope estimation.

## 1 Introduction

Radio-echo sounding (RES) is a well established technique for remotely measuring the thickness of ice sheets. The use of synthetic aperture radar (SAR) focusing improves gain and azimuth resolution of the echograms. Overall, state-of-the-art SAR processing offers information about the spatial properties of the ice sheet and the strength of the response, which is used to determine ice thickness, internal layers' orientation and bedrock conditions, i.e. presence or absence of water. There exist several SAR algorithms for focusing RES data, among them 1-D matched filtering (Legarsky et al., 2001), the $\omega - k$ migration (Leuschen et al., 2000), 2-D matched filtering (Heliere et al., 2007; Peters et al., 2007), and multilook time-domain back-projection (Mishra et al., 2016). Additionally, Holschuh et al. (2014) offer a method for improving SAR focusing of internal layers by introducing a correction of attenuation, migration and radial spreading for the returns from tilted internal layers.

Previous studies of angular backscattering properties of the ice sheet and bed are performed in (Jezek et al., 2009; Smith, 2014; Schroeder et al., 2013; MacGregor et al., 2015). Jezek et al. (2009) offer a technique for studying the backscattering properties of the ice sheet and bed using a special subaperture SAR approach. The authors study the dependency of the surface and bed return power on the incidence angle, the effect of the surface slope on the surface return power, they show that the response of the internal layers is specular, and propose incoherent presumming of subapertures to improve the signal-to-noise ratio (SNR) of internal layers. Smith (2014) estimates an optimal value for the SAR beamwidth based on the bedrock SNR. Schroeder et al. (2013) offer an approach for detecting the presence of subglacial water at the bed based on its angular

backscattering characteristics. The authors estimate the specularity of the bed returns by comparing power contributions in two HiCARS $60\,\mathrm{MHz}$ (Peters et al., 2007) SAR echograms with synthetic apertures of $700\,\mathrm{m}$ and $2000\,\mathrm{m}$. MacGregor et al. (2015) introduce two new methods for estimating the slope of internal layers, among them the Doppler centroid method, which uses the fact that internal layers' returns are highly specular. The authors use azimuth Fourier transform of short overlaping range-

5 compressed RES data blocks, and derive the slope of internal layers from the wavenumber of the corresponding Doppler centroids.

Mishra et al. (2016) introduce a novel approach for SAR processing of RES data, where the processing chain generates a number of SAR echograms, each corresponding to a particular incidence angle in along-track. A subset of the echograms with the highest SNR is then selected for further processing. The authors aim at improving the SNR of the weak bed echos in outlet

glaciers and perform no further analysis of the backscattering properties of the ice-sheet and bed.

In this paper we introduce a new flexible technique to analyze the angular backscattering properties of the ice-sheet and bed, which can be applied to previously conventionally SAR focused complex-valued echograms. Better understanding of those properties allows us to offer novel strategies for improving internal layer and possibly bed SNR, to mitigate the surface multiple return, and to train sparsyfing dictionaries for model-based cross-track focusing methods such as (Wu et al., 2011;

Heister and Scheiber, 2016).

This paper begins with a description of the employed SAR focusing algorithm for RES data in Sect. 2. After that we introduce the technique for analyzing angular backscattering properties of the ice-sheet and bed in Sect. 3. In Sect. 4 we analyze the processing results for two RES datatakes collected by the Center for Remote Control of Ice Sheets (CReSIS), Kansas, USA using their Multi-Channel Radar Depth Sounder (MCRDS) (Lohoefener, 2006; Marathe, 2008) during the Greenland campaign

in 2008 (CReSIS, 2012). Based on the results of Sect. 4, we discuss and demonstrate approaches for improving internal layer visibility and for mitigating the surface multiple in Sect. 5. Potential impacts for the scientific evaluation of SAR focused RES data with large beamwidth are discussed in Sect. 6. Finally, summary and conclusions are given.

## 2   SAR focusing

We perform SAR focusing of RES data using a modification of the range-Doppler algorithm. The processing is done in

overlapping azimuth blocks with each block processed as described in Algorithm 1. For each block we assume the platform to fly with a constant velocity $v$, the ice surface to have a constant along-track slope $\psi$, and the ice sheet to have a constant refractive index $n_{\mathrm{ice}} = 1.78$. We also assume that the electromagnetic wave propagation obeys Snell's law for a two-layer air-ice model shown in Fig. 1. The number of azimuth samples in each block is selected to satisfy at least twice the desired SAR beamwidth of $\Delta\theta = 30°$. We additionally assume that the azimuth antenna pattern is broad enough so that its variation

for incidence angles in the interval $\theta = [-15°, 15°]$ can be safely ignored.

We now describe the inputs for Algorithm 1 using the notation where $\tau$ denotes range time, $f_\tau$ denotes range frequency, $\eta$ denotes azimuth time, and $f_\eta$ denotes azimuth frequency.

Range compression, which is a signal processing technique for improving the radar range resolution, is implemented using a matched filter $H_{RC}$ equal to a complex conjugate of the Fourier transform of the transmitted signal weighted by Hamming or Blackman window for the side-lobe suppression.

The range-Doppler algorithm assumes a linear motion trajectory of the platform, therefore the motion compensation, a procedure that corrects the platform's trajectory deviation from a linear reference trajectory, is needed. We implement it using a filter $H_{MOCO}$, which only corrects for a vertical component of the platform's deviation from a reference track in the range frequency domain.

---

**Algorithm 1** SAR Focusing

**Require:** raw data DATA, filters $H_{RC}$, $H_{MOCO}$, and $H_{REF}$, amount of RCM $\Delta R_{RCM}$.
**Ensure:** SAR focused echogram $DATA_{SAR}$
1: $DATA := FFT_{range}(DATA)$
2: $DATA := DATA \cdot H_{RC} \cdot H_{MOCO}$
3: $DATA := IFFT_{range}(DATA)$
4: $DATA := FFT_{azimuth}(DATA)$
5: **for** $f_\eta \in [-B_{az}/2, B_{az}/2]$ **do**
6: $\quad DATA[:, f_\eta] := interp(DATA[:, f_\eta], \Delta R_{RCM}[:, f_\eta])$
7: **end for**
8: $DATA := DATA \cdot H_{REF}$
9: $DATA_{SAR} := IFFT_{azimuth}(DATA)$
10: **return** $DATA_{SAR}$

---

As the platform moves in azimuth, the response from a target spreads across multiple range bins. Range cell migration correction (RCMC) is an operation that removes this range variation, bringing the target response to a fixed range bin at every azimuth position. RCMC is performed in the range-Doppler domain. During RCMC every range line is shifted by the time corresponding to the amount of range cell migration $\Delta R_{RCM}$. The spatially variant shift in range is implemented using a sinc interpolator with Lanczos window ($a = 2$) and the length three times exceeding maximal $\Delta R_{RCM}$. Finally, azimuth compression is done by applying the $H_{REF}$ filter. We now derive equations for $\Delta R_{RCMC}$ and $H_{REF}$.

From Fig. 1 the optical path length $R$ from the radar at azimuth $x = \eta \cdot v$ to a point target at depth $d$ is

$$R(d, \eta) = R_{air} + n_{ice} R_{ice}, \tag{1}$$

where the geometric lengths that the electromagnetic wave travels in air and ice are

$$R_{air} = \sqrt{(R_0^2 + s \cdot \tan \psi)^2 + (x - s)^2}, \tag{2}$$

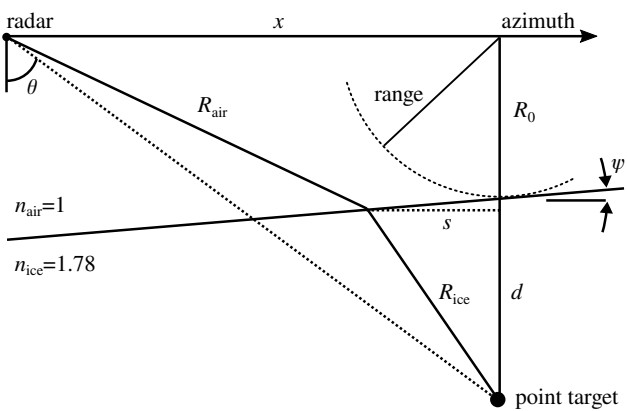

**Figure 1.** Along-track geometry.

$$R_{\text{ice}} = \sqrt{s^2 + (d - s \cdot \tan\psi)^2}. \tag{3}$$

Both (Eq. 2) and (Eq. 3) depend on an unknown location of the refraction point $s$, which is a function of time $\eta$ and depth $d$. The refraction point $s$ can be found by solving a fourth-order polynomial equation (Heliere et al., 2007; Scheiber et al., 2008) or, more efficiently, by using Newton's optimization method, which iteratively finds $s$ that minimizes (Eq. 1) with the following update rule at $(i+1)$-th iteration

$$s_{i+1} = s_i - R'(s)/R''(s), \tag{4}$$

where we initialize the refraction point with $s_0 = 0$.

Knowing $s$ we calculate the phase shift of the received signal with respect to the time $\eta = 0$ when the platform crosses the origin of the $x$-axis

$$\Delta\phi(\eta) = \frac{4\pi}{\lambda_0}\Delta R(\eta) = \frac{4\pi}{\lambda_0}(R(\eta) - R(0)), \tag{5}$$

where $\lambda_0$ is the wavelength of the transmitted wave in the air.

The Doppler frequency shift of the received signal is proportional to the derivative of Eq. (5) in time

$$\Delta f(\eta) = \frac{1}{2\pi}\frac{\partial\phi(\eta)}{\partial\eta}. \tag{6}$$

Knowing Eq. (6) for each depth $d$ and azimuth position $x$, we compute the amount of range cell migration in the range-Doppler domain $\Delta R_{\text{RCM}}(d, f_\eta)$ by interpolating its time domain counterpart $\Delta R_{\text{RCM}}(d, \eta) = \tau \cdot c_0/2 - \Delta R(d, \eta)$ onto a regularly sampled azimuth frequency grid $f_\eta \in [-B_{\text{az}}/2, B_{\text{az}}/2]$, where $f_\eta = \pm B_{\text{az}}/2$ corresponds to incidence angles $\theta = \pm 15°$.

Finally, we compute $\Delta\phi(f_\eta)$ by interpolating Eq. (5) onto $f_\eta$, and calculate the matched filter for SAR focusing $\mathrm{H_{REF}}$ as

$$\mathrm{H_{REF}}(\tau, f_\eta) = \exp\left(-j\Delta\phi(f_\eta)\right). \tag{7}$$

We note that more precise and less restrictive SAR focusing algorithms for ice-sounder data exist, such as time-domain back-projection (Mishra et al., 2016); our choice of a particular approach described above is based on simplicity of implementation and its sufficiency for the subsequent analysis of the ice-sheet and bed angular backscattering properties.

## 3 Multiple subbands processing

In order to analyze the dependency of the backscattering properties of the ice sheet and bed on the incidence angle, we divide the azimuth spectrum of an echogram into $N$ overlapping subbands of beamwidth $\Delta\theta_{\mathrm{sub}} = 2°$ and an overlap between two adjacent subbands of $1°$, with each subband weighted by a rectangular window. The central frequency of the $n \in (1, N)$ subband, $f_0(\theta_n)$, corresponds to the incidence angle of interest $\theta_n \in [-14°, 14°]$

$$f_0(\theta_n) = \frac{2v\sin\theta_n}{\lambda_0}. \tag{8}$$

Each subband is then accordingly zero-padded in azimuth so that all $N$ subbands have the same size. After that an inverse azimuth Fourier transform is applied to each subband to get a set of $N$ echograms $I_n$, each containing returns coming predominantly from the corresponding incidence angle $\theta_n$.

The positions of ice-sheet features of interest, such as surface, internal layers and bed, are then manually selected from an echogram $I_{\mathrm{incoh}}$ calculated as the incoherent sum

$$I_{\mathrm{incoh}} = \sum_{n=1}^{N} |I_n|. \tag{9}$$

We note that the ice sheet features can be tracked automatically, however, for the small amount of data we analyse in the paper, manual selection is feasible.

## 4 Greenland MCRDS data

We apply the approach presented in Sect. 3 to RES data collected by CReSIS (CReSIS, 2012) using their MCRDS system (Lohoefener, 2006; Marathe, 2008). The main parameters of the radar and the acquisitions are summarized in Table 1. Two chirps with different durations were transmitted alternately on a pulse-to-pulse basis, with a $3\,\mu\mathrm{s}$ chirp intended to capture the surface and the shallow internal layer returns (shallow mode), and a $10\,\mu\mathrm{s}$ chirp intended to capture deeper internal layers and bed returns (deep mode). We employ the availability of multiple cross-track channels of MCRDS to increase the SNR of nadir returns by combining the SAR echograms of the cross-track channels using conventional delay and sum beamforming.

**Table 1.** Parameters of MCRDS acquisitions.

| Parameters | track 1 | track 2 |
|---|---|---|
| Central frequency | 150 MHz | |
| Chirp bandwidth | 20 MHz | |
| Chirp duration | $3/10\,\mu s$ | |
| Sampling frequency | 120 MHz | |
| Effective PRF | 78 Hz | 156 Hz |
| Number of cross-track channels | 16 | 6 |
| Effective cross-track aperture | 14.34 m | 4.79 m |
| Acquisition date | 2008-07-20 | 2008-08-01 |
| Acquisition start UTC | 18:32:29 | 16:49:49 |
| Acquisition end UTC | 18:47:33 | 17:07:23 |
| Average height over surface | 160 m | 800 m |
| Average velocity | $78\,\mathrm{m\,s^{-1}}$ | $65\,\mathrm{m\,s^{-1}}$ |

We select two tracks, both flown over Greenland in summer 2008, both approximately 70 km long. The track flown from the inland towards Jakobshavn glacier is referred to as track 1, the track flown over Southeast Greenland in a North-East direction is referred to as track 2. The regions of interest, their topography and flight trajectories are shown in Fig. 2. These particular datatakes are chosen to demonstrate how different bed topography affects the reflective properties of the internal layers; the

5    bed in track 1 has depth varying in the interval $d_\mathrm{bed} \in [2170\,\mathrm{m}, 3030\,\mathrm{m}]$ and slopes varying in the interval $\psi_\mathrm{bed} \in [-35°, 33°]$, the corresponding intervals for track 2 are $d_\mathrm{bed} \in [640\,\mathrm{m}, 1970\,\mathrm{m}]$ and $\psi_\mathrm{bed} \in [-62°, 65°]$. Here we calculate the slopes of the bed and the internal layers as

$$\psi_\mathrm{bed/layer}(x) = \tan^{-1}\left(\frac{\partial d_\mathrm{bed/layer}(x)}{\partial x} \cdot n_\mathrm{ice}\right), \tag{10}$$

where $n_{ice}$ scales the geometric slope to correspond to the incidence angle observed by the radar.

10    The full bandwidth echogram of track 1 is shown in Fig. 3(a). To produce the figure we combine echograms of the shallow and deep modes, rebin the echogram in azimuth by a factor of 8, and add a depth-dependent amplitude ramp of $1.5\,\mathrm{dB}/100\,\mathrm{m}$ to improve visibility of the deep internal layers and the bed. The internal layers are visible until $d \approx 2\,\mathrm{km}$. The gaps in internal layer visibility occur at azimuth positions where the bed slope is the steepest.

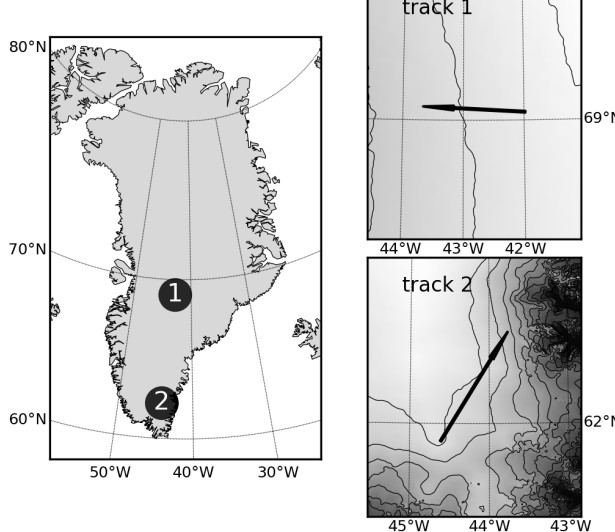

**Figure 2.** MCRDS datatakes on a map. The map of Greenland is plotted using a stereographic projection with a central meridian of $41^\circ$ W and a central parallel of $72^\circ$ N. Isolines on maps correspond to a surface elevation change of $250\,\text{m}$.

First, we investigate reflective properties of the ice surface. Figure 3(b) shows the normalized reflectivity power of the surface as a function of incidence angle. The surface response is specular, with the incidence angle corresponding to the maximum intensity $\theta_{\max(I)}$ varying slowly in azimuth.

To study backscattering properties of internal layers we select a single internal layer with depth $d \approx 1870\,\text{m}$ at the azimuth
position $x = 0\,\text{km}$. A deep layer is selected in order to avoid undesired contributions of the off-nadir surface returns. Figure 3(c) shows the internal layer's normalized power together with its slope, computed from (Eq. 10), drawn as a white line. We use bicubic interpolation to plot the figure. The internal layer response is specular, with $\theta_{\max(I)}$ proportional to the layer's geometric slope.

A further insight into the behavior of the internal layer's response is given in Fig. 3(d), where for each pixel of $I_{\text{incoh}}$ we
colorcoded the incidence angle corresponding to the maximum intensity $\theta_{\max(I)}$ . Prior to plotting, we additionally applied a median filter of size $(5, 5)$ and bicubic interpolation. The black lines on the figure correspond to the surface and bed return positions. The figure shows correlation between $\theta_{\max(I)}$ and the bed slope, with the blue and the red color appearing at azimuth positions with negative and positive bed slope correspondingly. Moreover, for a given azimuth position $x_0$ the absolute value of $\theta_{\max(I)}$ increases with depth, therefore, according to Fig. 3(c), the absolute value of internal layers' slope also increases with
depth. This implies that the deeper the internal layer is located, the more its shape resembles the shape of the bed.

Figure 3(e) shows the normalized power of the bed response, where, prior to the normalization, we additionally compensate for the two-way propagation power loss of $2\,\text{dB}/100\,\text{m}$. The incidence angle $\theta_{\max(I)}$ of the bed response varies in azimuth, overall the response is wide, meaning the bed is a rough surface for a radar with $\lambda_0 = 2\,\text{m}$.

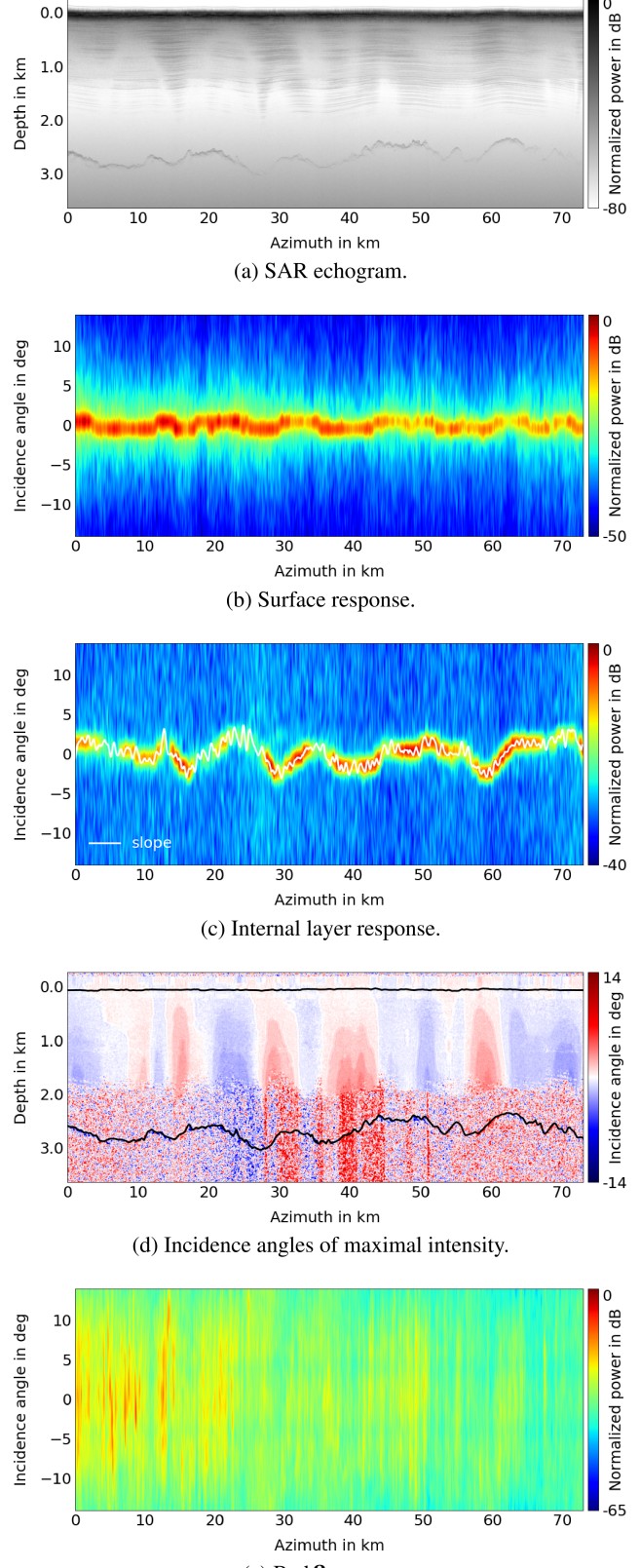

(a) SAR echogram.

(b) Surface response.

(c) Internal layer response.

(d) Incidence angles of maximal intensity.

(e) Bed response.

**Figure 3.** Backscattering characteristics of the ice-sheet and bed for track 1.

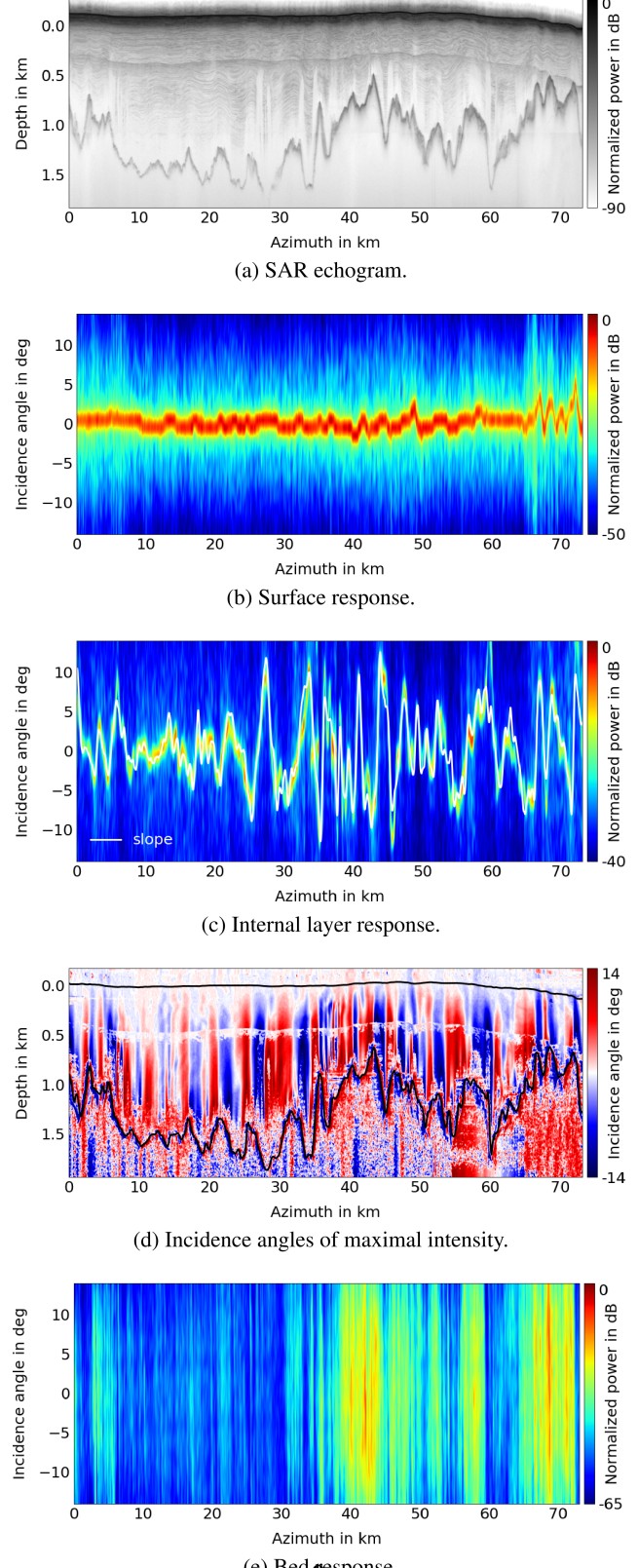

(a) SAR echogram.

(b) Surface response.

(c) Internal layer response.

(d) Incidence angles of maximal intensity.

(e) Bed response.

**Figure 4.** Backscattering characteristics of the ice-sheet and bed for track 2.

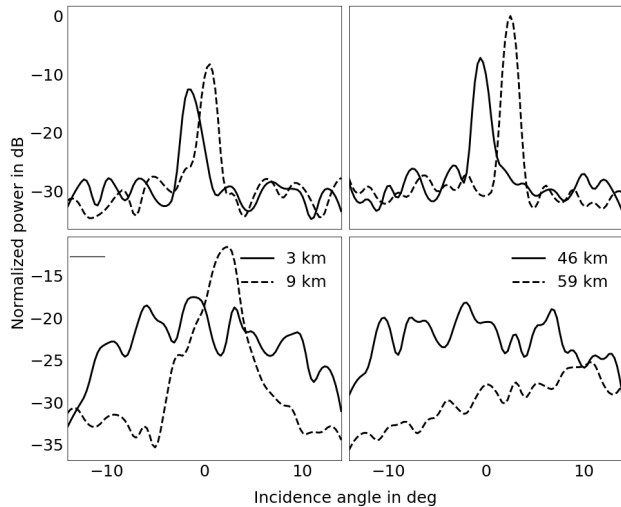

**Figure 5.** Internal layer and bed returns for track 1.

Figure 5 shows the dependency of the return power of the internal layer previously selected for Fig. 3(c) and bed for the fixed azimuth positions $x = (3, 9, 46, 59)\,\mathrm{km}$. Those particular azimuth positions are selected to demonstrate the variety of shapes of reflective signatures for the bed and the persistent signature shape for the internal layer. We use quadratic interpolation to smooth the signatures.

The full bandwidth echogram for track 2 is shown in Fig. 4(a), where we add a depth-dependent amplitude ramp of $2\,\mathrm{dB}/100\,\mathrm{m}$. Here the bed topography varies stronger as compared to track 1, the internal layers are visible close to the bed with gaps appearing at azimuth positions where the absolute value of bed slope is the highest; the surface multiple is also present in the echogram.

Figure 4(a) shows the normalized reflectivity power of the surface. The surface response is similar to the one for track 1,
with higher variation of $\theta_{\max(I)}$ occurring starting from azimuth $x = 65\,\mathrm{km}$.

Reflective properties of a single internal layer with depth $d \approx 440\,\mathrm{m}$ at the azimuth position $x = 0\,\mathrm{km}$ are shown in Fig. 4(b). Here we select a shallow layer because $\theta_{\max(I)}$ for deeper layers would lie outside the interval $\theta_n \in [-14°, 14°]$ previously selected in Sect. 3. The incidence angle $\theta_{\max(I)}$ in Fig. 4(c) varies stronger and more frequently as compared to that in Fig. 3(c).

Figure 4(d) is plotted similarly to Fig. 3(d). As expected, we observe larger color gradients for internal layers for track 2,
whereas incidence angles of the surface multiple lie around $\theta_n \approx 0°$ in white, corresponding to the ice surface.

The normalized power of the bed response for track 2 is shown in Fig. 4(e).

In Fig. 5 we compare the responses of the previously selected internal layer and the bed for the fixed azimuth positions $x = (3, 25, 32, 54)\,\mathrm{km}$. We again observe specular reflections from the internal layer and wider reflections from the bed, with $\theta_{\max(I)}$ for the bed and the internal layer positively correlated for each selected position.

For both tracks the mean value of the surface and the layer beamwidth at $-6\,\mathrm{dB}$ level is $2.2°$. The shape of the bed response varies and does not necessarily have a prominent single peak, therefore we do not calculate its beamwidth. We suggest to use

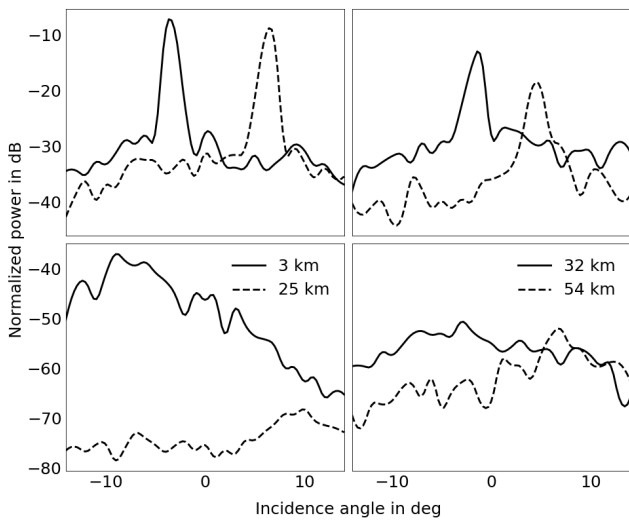

**Figure 6.** Internal layer and bed returns for track 2.

the variance of the bed angular response to quantify its spread when the bed roughness or the presence of basal water is of interest. For more details we refer the readers to Sect. 6.

## 5 Enhancement of SAR echograms

In this section we offer two straightforward applications of the results provided in Sect. 4 for improvement of SAR focused RES data.

First, the fact that an internal layer response is narrow means that for a given depth and azimuth it contributes only to a limited azimuth frequency range of a SAR echogram spectrum. For the azimuth spectrum of a small block of a SAR echogram we see that at each depth internal layer contribution is clustered around the frequency corresponding to the layer slope, which is demonstrated in Fig. 7. This allows us to use spectral filtering to improve the SNR of the internal layers. We perform the filtering using $250\,\mathrm{m}$ azimuth blocks with a $70\%$ overlap. We apply a Fourier transform in azimuth to each block, after that for each depth we select a frequency corresponding to the maximum spectrum intensity $f_{\mathrm{max}(I)}(d)$ and fit it into a piecewise linear regression with $n_{\mathrm{pieces}} = 3$ to make the estimate $\hat{f}_{\mathrm{layer}}(d)$ of the internal layers' frequency $f_{\mathrm{layer}}(d)$ more robust against outliers. Depending on the shape of $f_{\mathrm{layer}}(d)$ other values for $n_{\mathrm{pieces}}$ as well as other types of regression (e.g. polynomial regression) might be used to calculate $\hat{f}_{\mathrm{layer}}(d)$. After that we nullify the part of the spectrum at frequencies lying outside the interval $\hat{f}_{\mathrm{layer}}(d) \pm 0.05\Delta B_{\mathrm{az}}$, which is marked with white lines in Fig. 7, whereas the part of the spectrum lying inside the interval is kept intact. Finally we apply an inverse Fourier transform in azimuth to each block and re-assemble the overlapping blocks. We apply this method to the track 1 SAR echogram and the results are shown in Fig. 8, where subsets of the SAR echogram before and after the processing are shown at the top and bottom correspondingly.

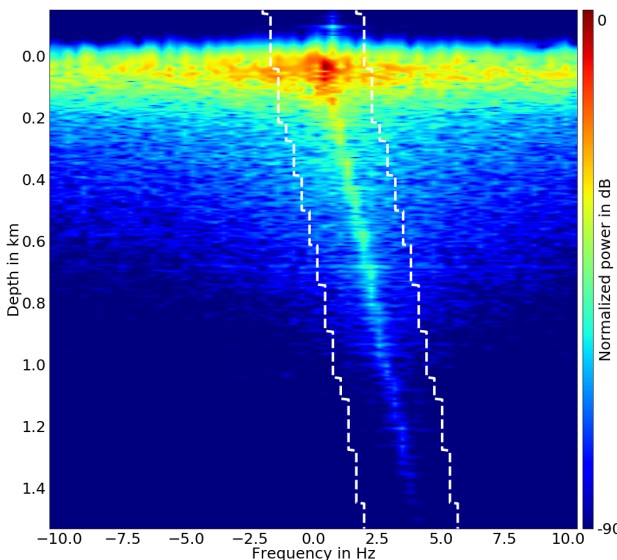

**Figure 7.** Azimuth spectrum of a $250\,\mathrm{m}$-long SAR block in track 1, azimuth position $x = 59\,\mathrm{km}$.

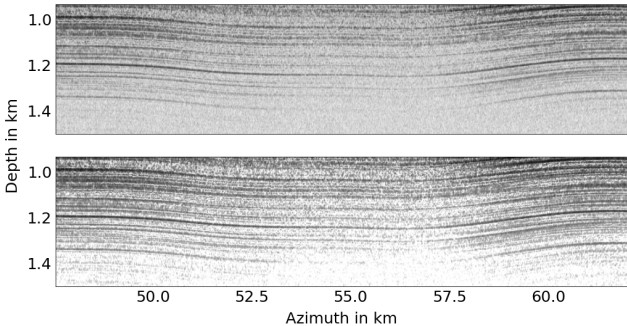

**Figure 8.** Improvement of internal layer visibility for track 1.

The procedure results in a $21.8\%$ sharpness improvement in terms of intensity squared metric (Fienup and Miller, 2003) (Eq. 11) with the mean intensity of the echograms normalized prior to the comparison.

$$s(I) = \sum_{i,j} I_{i,j} \cdot I_{i,j}^*. \tag{11}$$

Figure 9 depicts the antenna power pattern, the layer power spectrum centered around $f_{\text{layer}}$, and the power spectrum of the noise. The noise power is proportional to the integral of its power spectrum. Prior to the spectral filtering this frequency range contains the entire processed bandwidth $\Delta B_{\text{az}}$, whereas after the filtering the interval is $f_\eta \in [f_{\text{layer}} - 0.05\Delta B_{\text{az}}, f_{\text{layer}} +$

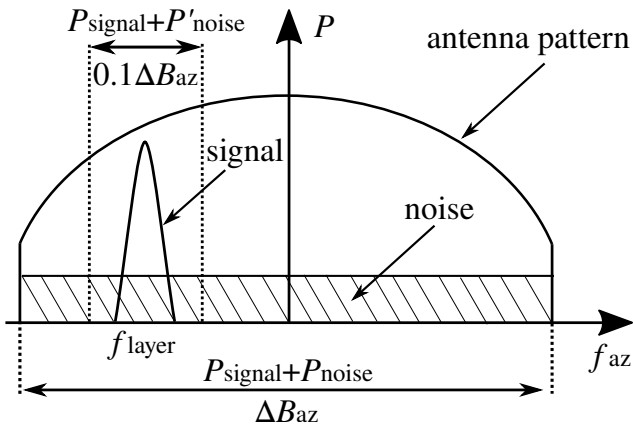

**Figure 9.** Noise reduction for internal layer SNR improvement.

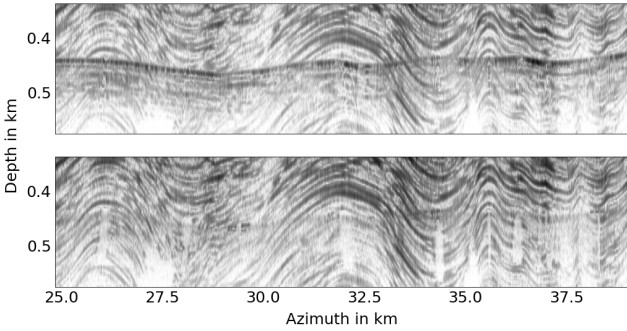

**Figure 10.** Mitigation of the surface multiple for track 2.

$0.05\Delta B_{\mathrm{az}}$]. Assuming white Gaussian noise, and that the signal energy is not affected by the filtering, the SNR improvement is $10\,\mathrm{dB}$.

Second, according to the Fig. 4(d), the contribution of the surface multiple return, which is a multipath reflection from the ice surface as well as from the upper internal layers and the bottom of the aircraft fuselage and wings, can be mitigated by
5   identifying and filtering out its contribution in the azimuth frequency domain, therefore revealing previously masked internal layers. This approach however only works in areas where the $\theta_{\max(I)}$ for internal layers and the surface multiple differ.

To demonstrate the approach we process a subset of a SAR echogram of track 2. The processing is done in blocks with similar parameters as for internal layer visibility improvement. In each block we locate the strongest surface multiple contribution at a depth $d_{\mathrm{mult}}$ corresponding to the doubled height of the aircraft over the ice surface, and at azimuth frequency $f_{\mathrm{mult}}$ which
10  corresponds to the frequency of strongest surface return. After that in each block we apply a 2-D notch filter located at depth $d \in [d_{\mathrm{mult}}, d_{\mathrm{mult}} + 100\,\mathrm{m}]$ and at frequency $f_{\eta} \in [f_{\mathrm{mult}} - 0.05\Delta B_{\mathrm{az}}, f_{\mathrm{mult}} + 0.05\Delta B_{\mathrm{az}}]$. The results are shown in Fig. 10.

# 6 Scientific utility of large beamwidth SAR processing

In this section we discuss the practical benefit of large beamwidth SAR processing of RES data for two scientific applications, namely basal water detection and estimation of internal layer slope.

Presence of water bodies can be detected in RES data using either amplitude or angular information. The amplitude detection is based on the fact, that the basal water produces stronger reflection compared to the grounded bed. However this method is prone to errors (Matsuoka, 2011) as the radar attenuation depends on the chemical composition of the ice as well as its temperature (MacGregor et al., 2007). The analysis of the angular backscattering distribution of the bed return on the other hand is free of the aforementioned limitations, and a specular bed response indicates the presence of basal water.

In order to quantify specularity of the bed return Schroeder et al. (2013) introduce the specularity content, a measure which is calculated as a ratio of energies of the bed return in two SAR echograms $I_1$ and $I_2$, focused with synthetic apertures $L_1 = 700\,\mathrm{m}$ and $L_2 = 2000\,\mathrm{m}$ correspondingly. The echogram $I_1$ contains specular returns, whereas the echogram $I_2$ contains both specular and non-specular returns. For a typical height above the ice surface $R_0 = 500\,\mathrm{m}$ and ice thickness $d = 2\,\mathrm{km}$ used in the survey (Schroeder et al., 2014), $L_1$ and $L_2$ respectively correspond to beamwidths $\Delta\theta_1 \approx 10°$ and $\Delta\theta_2 \approx 28°$.

We compare the specularity content used so far with the variance of angular backscattering of the bed, which is a measure of the bed angular distribution spread, introduced in Sect. 4. We calculate the specularity content using $\Delta\theta_1 = 10°$ and $\Delta\theta_2 = 30°$. To calculate the variance we normalize the energy of the angular backscattering in each azimuth position. For comparison we take two parts of track 2 each $2.3\,\mathrm{km}$ long. Figure 11 shows the angular distributions of the bed return energy, normalized with respect to the highest value of the distribution at each azimuth position, the area inside the dashed horizontal lines corresponds to $\Delta\theta_1$. The specularity content is low at azimuth positions A and D, failing to detect specular bed reflections located outside of $\Delta\theta_1$; the specularity content is high at azimuth positions B and C, but decays rapidly as the bed reflection moves outside of $\Delta\theta_1$. The variance on the other hand is insensitive to such angular shift of the reflected energy, therefore making it possible to detect additional specular reflections. Using the variance instead of the specularity content can potentially lead to a better detection and mapping of subglacial water bodies, especially in areas where the bed is tilted in azimuth.

Internal layers, which are generally attributed to changes of electrical properties of the ice sheet, are frequently observed in RES echograms. Because the layers are believed to be isochrones (Bogorodsky et al., 1985; Jacobel and Hodge, 1995), when tracked in RES data, they provide information about changes in the ice flow (Siegert, 2004) and snow accumulation rate in the past (Fahnestock et al., 2001). The availability of a large amount of RES data and the fact that the manual layer tracking is prohibitively expensive (Sime et al., 2011) motivate the development of automated layer tracking algorithms. Some of the tracking algorithms use reflection slope of the layers as their input, with the slope integration in azimuth producing a synthetic isochrone.

MacGregor et al. (2015) introduce two new methods for the slope estimation, namely horizontal phase gradient and Doppler centroid methods. Both methods use coherent RES data (phase preserved). As stated in (MacGregor et al., 2015) currently the data have been range compressed, but without SAR focusing. The use of range compressed as opposed to SAR focused RES data might lead to erroneous estimation of the slope due to the displacement of internal layers in azimuth. The displacement is

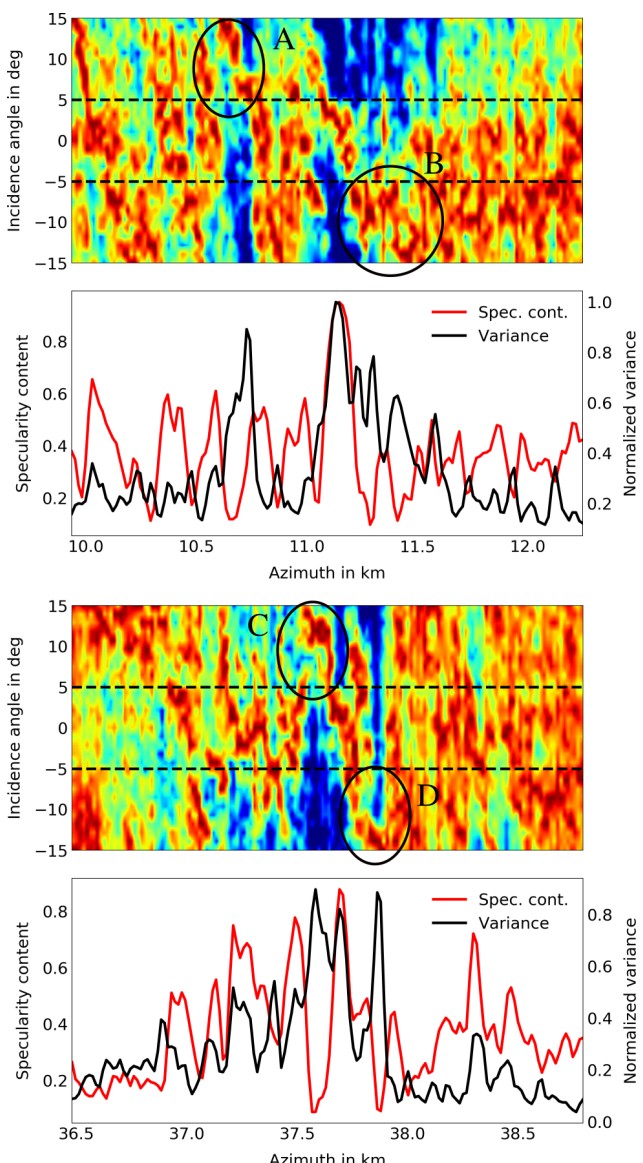

**Figure 11.** Specularity content and variance as measures of specularity of the bed return for track 2.

due to the fact that, the internal layer reflection is specular, which in turn means that prior to SAR focusing the return from an internal layer appears at the azimuth position where the incident energy is normal to the layer's surface. SAR focusing registers the return at its zero Doppler position, which corresponds to the nadir direction.

Figure 12 schematically illustrates this effect with two examples. For the sake of simplicity we ignore the ray bending due to the difference in refractive properties of the air and ice. The azimuth positions at which the layer returns are registered in range compressed and SAR focused data are marked with circles and squares, respectively. A convex internal layer in Fig. 12 (a)

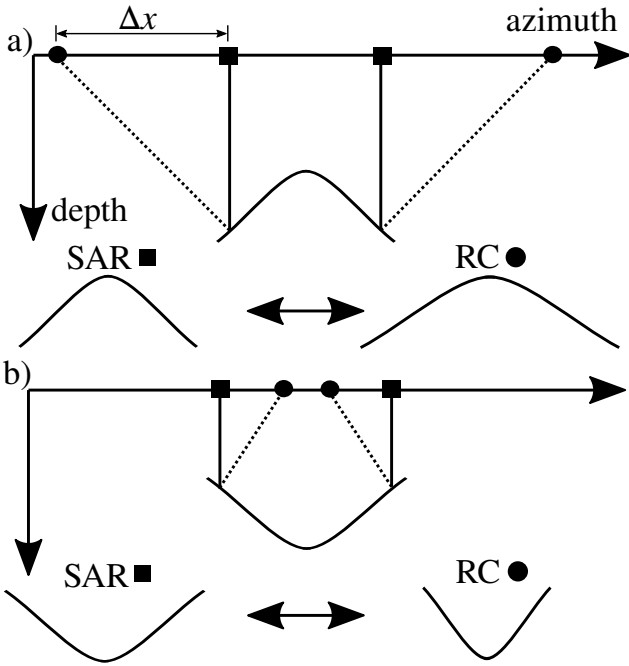

**Figure 12.** Azimuth displacement of a tilted internal layer in range-compressed data.

appears stretched in azimuth in range compressed data, whereas a concave internal layer in Fig. 12 (b) appears shrunk (in extreme cases reflections from the left and the right would be overlaid).

The amount of the displacement $\Delta x$ depends on the layer's geometric slope $\alpha$, its depth $d$, platform height above the surface $R_0$, the refractive index of the ice $n_{\text{ice}}$, and the surface slope $\psi$. When $\psi = 0°$, the displacement is calculated as

$$\Delta x = R_0 \tan(\sin^{-1}(n_{\text{ice}} \sin \alpha)) + n_{\text{ice}} d \tan \alpha. \tag{12}$$

Figure 13 shows the dependency of $\Delta x$ upon the layer geometric slope $\alpha$ and depth $d$ when the height over the surface is $R_0 = 800\,\text{m}$, the ice refractive index is $n_{\text{ice}} = 1.78$, and $\psi = 0°$.

Real data examples for both cases are demonstrated in Fig. 14, where we compare range compressed and SAR focused subset of RES data for track 2. For the range compressed data, the internal layers appear stretched in azimuth in areas A and C, whereas for the area B the layers shrink and even overlay at the lower depth.

## 7 Summary and conclusions

In this paper we offered a new approach to study scattering characteristics of ice-sheets which is based on the division of a conventionally focused large beamwidth ice sounder SAR echogram into a set of subbands each of which corresponds to a particular incidence angle in along-track direction. We estimated and compared scattering characteristics of the ice surface, in-

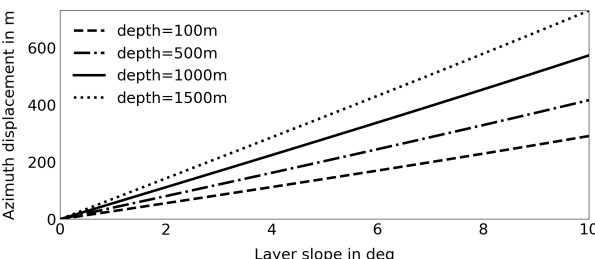

**Figure 13.** Internal layer displacement in azimuth.

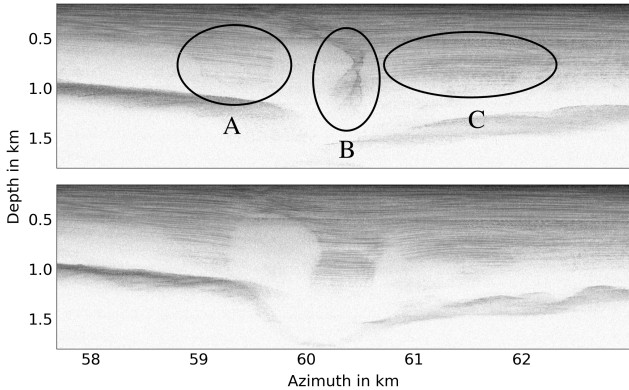

**Figure 14.** Stretching and shrinkage of internal layers in range-compressed data of track 2. The range compressed echogram is at the top, the SAR focused echogram is at the bottom.

ternal layers and bed for two datatakes in Greenland. For those datatakes the surface and internal layers have narrow responses, which corresponds to a smooth specular surface, while the bed response is wide, which corresponds to a rough surface. The scattering properties carry information which can be used to estimate characteristics of the bed roughness (Fung and Eom, 1983), with the specular bed response indicating the likely presence of subglacial water at the bed (Schroeder et al., 2013).

5     Based on the scattering characteristics of internal layers, we offered a post-processing technique to improve their visibility. By taking a small azimuth block of a SAR echogram, within which the orientation of internal layers varies slightly in along-track, we observe that the internal layer's contribution to the block's azimuth spectrum is sparse and is clustered around the frequency corresponding to the internal layer's slope. This observation directly suggests a way to improve internal layers' SNR by keeping only those spectral components where the internal layers' contributions are present. This post-processing technique

10   can improve spatial tracking and interpretation of both horizontal and tilted internal layers. As a subject for further studies we suggest that denoising of all ice-sheet features in a SAR echogram is possible by finding a sparse representation of the echogram given a sparsyfing dictionary learned on patches with high SNR.

We also demonstrated a way to reduce the undesired contribution of the surface multiple return, which masks internal layers on corresponding depths. The reduction is possible whenever the surface multiple and the masked layer contributions come from different incidence angles, in which case they are separable in the azimuth frequency domain.

Finally, we discussed the potential benefit offered by the analysis of RES data focused using large beamwidth SAR with regards to the bed specularity characterization and for the correct azimuth positioning of the tilted internal layers.

*Competing interests.* The authors declare that they have no conflict of interests.

*Acknowledgements.* We acknowledge the use of data from CReSIS generated with support from the University of Kansas, NASA Operation IceBridge grant NNX16AH54G, and NSF grant ACI-1443054. The authors would also like to thank John Paden of CReSIS for answering the sensor and data related questions.

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
