# Peer review of "Benefits of Coherent Large Beamwidth Processing of Radio-Echo Sounding Data"

_The Cryosphere, 2018_

## Referee Comment (RC1) · Anonymous Referee #1 · 8 May 2018

**Review Comments**

| Title: | Benefits of Coherent Large Beamwidth Processing of Radio-Echo Sounding Data |
|---|---|

**Journal:** The Cryosphere
**Date:** 05/08/2018

**General Comments:**

In this manuscript, the authors present a novel radar focusing method that improves image quality based on the scattering properties of the ice surface, englacial reflectors, and ice bottom. This method can be used to improve the signal to noise ratio of englacial reflectors, can aid in the elimination of surface clutter and surface multiples, and can be used to diagnose the roughness characteristics of the surface and subsurface reflectors. The authors present the algorithm in sufficient detail to reproduce the results of their analysis, but the discussion of the *scientific* impact of the algorithm is limited -- at present, this work is more appropriate for IEEE. To publish this work in The Cryosphere, the authors should include either (a) substantial discussion of the scientific utility of the new algorithm, or (b) more rigorous interpretation of their test data.

**Specific Comments:**

The aims and scope of The Cryosphere sets standards for originality and impact of the published works. These are areas of weakness for the manuscript at present. Direction of arrival analysis is already well established in the literature (Al-Ibadi et al., 2017; Jezek et al., 2009; Wang et al., 2016). The authors do present a novel numerical scheme for inferring direction of arrival in the along-track direction. But their primary analysis is to show that englacial layers are more specular than the bed, and that deep layers are more conformal with the bed than shallow ones, two observations already discussed widely in the literature.

There are few issues with the information presented, which makes this manuscript challenging to review. The authors cite (but do not pursue) two possible scientific applications of their method – (1) to characterize bed roughness or subglacial hydrology (Schroeder et al., 2013) and (2) to better estimate layer slopes (Macgregor et al., 2015) and potentially relate those slopes to ice dynamics. To make this manuscript more appropriate in The Cryosphere, the authors could provide more substance in a number of ways: discuss the ice dynamic understanding derived from resolving steep englacial layers (Hindmarsh et al., 2006; Holschuh et al., 2017), spatially constrain the roughness characteristics in their test data and relate those to the underlying geology or ice flow behavior (Schroeder et al., 2014), or work toward a better understand the waveform characteristics of englacial reflectors (Drews et al., 2012) or diffuse scatterers (Jordan et al., 2017). Right now, the manuscript simply defines an algorithm and applies it – there is essentially no interpretation of results. While the summary and conclusions might be interesting to radioglaciologists who regularly process radar data, they are unlikely to be interesting to radar-data end users or a general Cryosphere audience.

**Technical Corrections:**

| Line # | Comment |
|--------|---------|
| Pg1, L10 | "The use of synthetic aperture radar (SAR) allows to improve…" – allows is missing an object. Allows who/what to improve? |
| Pg1, L13 | "Several algorithms were offered for …" – offered by whom? Offered for what? Odd phrasing. |
| Pg2, L3 | "MacGregor et al. (2015) introduce two new methods for estimating the slope of internal layers, among them the Doppler centroid method, which uses the fact, that internal layers' returns …" – remove comma after fact. |
| Pg2, L22 – P3, L4 | Section 2 is written with an electrical engineering audience in mind – clearer definitions are needed to make this accessible in The Cryosphere. This includes your coordinate system (maybe include "azimuth" and "range" in figure 2?), pulse compression, motion compensation and range cell migration corrections. |
| P4, Eq 2 | Your variable definitions and nomenclature in this section is confusing. Figure 1 defines the height above the surface as R0 and the depth below surface as d, but then your height above the surface is h in eq 2? h is actually never defined. |
| Pg11. L10 | "… and fit it into a piecewise linear regression …" -- this is not clear enough to be reproduced, but seems critical to your image refinement technique. |

**Review References:**

Al-Ibadi, M., Sprick, J., Athinarapu, S., Stumpf, T., Paden, J., Leuschen, C., … Van Wychen, W. (2017). DEM extraction of the basal topography of the Canadian archipelago ICE caps via 2D automated layer-tracker. *International Geoscience and Remote Sensing Symposium (IGARSS)*, *2017–July*, 965–968. https://doi.org/10.1109/IGARSS.2017.8127114

Drews, R., Eisen, O., Steinhage, D., Weikusat, I., Kipfstuhl, S., & Wilhelms, F. (2012). Potential mechanisms for anisotropy in ice-penetrating radar data. *Journal of Glaciology*, *58*(209), 613–624. https://doi.org/10.3189/2012JoG11J114

Hindmarsh, R. C. A., Leysinger Vieli, G. J. M. C., Raymond, M. J., & Gudmundsson, G. H. (2006). Draping or overriding: The effect of horizontal stress gradients on internal layer architecture in ice sheets. *Journal of Geophysical Research: Earth Surface*, *111*(2). https://doi.org/10.1029/2005JF000309

Holschuh, N., Parizek, B. R., Alley, R. B. R. B., & Anandakrishnan, S. (2017). Decoding ice sheet behavior using englacial layer slopes. *Geophysical Research Letters*, *44*(11), 5561–5570. https://doi.org/doi:10.1002/2017GL073417

Jezek, K., Gogineni, P., Wu, X., Rodriguez, E., Rodriguez, F., & Freeman, A. (2009). *Global ice sheet mapping observatory: Airborne experiments. Global Ice Sheet Mapping Observatory project - Final Report*. https://doi.org/10.1109/RADAR.2009.4976977

Jordan, T. M., Cooper, M. A., Schroeder, D. M., Williams, C. N., Paden, J. D., Siegert, M. J., & Bamber, J. L. (2017). Self-affine subglacial roughness: consequences for radar scattering and basal thaw discrimination in northern Greenland. *The Cryosphere*, *11*, 1247–1264. https://doi.org/10.5194/tc-2016-283

Macgregor, J. a, Fahnestock, M. a, Catania, G. a, Paden, J. D., Gogineni, S. P., Young, S. K., … Morlighem, M. (2015). Radiostratigraphy and age structure of the Greenland Ice Sheet. *Journal of Geophysical Research: Earth Surface*, *120*(2), 212–241. https://doi.org/10.1002/2014JF003215.Received

Schroeder, D. M., Blankenship, D. D., Young, D. A., Witus, A. E., & Anderson, J. B. (2014). Airborne radar sounding evidence for deformable sediments and outcropping bedrock beneath Thwaites Glacier, West Antarctica. *Geophysical Research Letters*, *41*(20), 7200–7208. https://doi.org/10.1002/2014GL061645

Schroeder, D. M., Blankenship, D. D., & Young, D. a. (2013). Evidence for a water system transition beneath Thwaites Glacier, West Antarctica. *Proceedings of the National Academy of Sciences of the United States of America*, *110*(30), 12225–8. https://doi.org/10.1073/pnas.1302828110

Wang, Z., Gogineni, S., Rodriguez-Morales, F., Yan, J. B., Paden, J., Leuschen, C., … Braaten, D. (2016). Multichannel Wideband Synthetic Aperture Radar for Ice Sheet Remote Sensing: Development and the First Deployment in Antarctica. *IEEE Journal of Selected Topics in Applied Earth Observations and Remote Sensing*, *9*(3), 980–993. https://doi.org/10.1109/JSTARS.2015.2403611

---

## Referee Comment (RC2) · J. Paden (Referee) · 14 May 2018

Title: Benefits of Coherent Large Beamwidth Processing of Radio-Echo Sounding Data
Authors: Anton Heister and Rolf Scheiber

The paper discusses a range Doppler algorithm for ice sheet synthetic aperture radar (SAR) processing and also discusses along-track angular characteristics of the scattering. The method is used to show the angular or Doppler spectrum of the surface, internal layers, and ice bottom, the slope field for the internal layers, and Doppler domain filtering of internal layers which also can suppress the surface multiple when the superimposed internal layers and surface multiple have different Doppler frequencies.

[Figure]

The paper is very well written with some nice examples. Even though this property of ice layers and SAR processing is well known in the ice radar community, a nice description like this is missing from the literature and the adaptation of the range-Doppler algorithm has not been published although others have done it before. The one issue is that this is a methodology paper and does not address any scientific question directly. On the other hand, the method described could be used to do so.

All comments and suggestions for improvement are included in the paper.

The most important points are to fill in descriptive information about the algorithm and to add/update some of the references.

Please also note the supplement to this comment:
https://www.the-cryosphere-discuss.net/tc-2018-61/tc-2018-61-RC2-supplement.pdf

**Supplement:**

**Benefits of Coherent Large Beamwidth Processing of Radio-Echo Sounding Data**

Anton Heister[1] and Rolf Scheiber[1]

[1]German Aerospace Center (DLR), Microwaves and Radar Institute, Wessling, Germany

**Correspondence:** Anton Heister (anton.heister@dlr.de)

**Abstract.** Coherent processing of radio echo sounding data of polar ice sheets is known to provide indication of bedrock properties and detection of internal layers. We investigate the benefits of coherent processing of a large azimuth beamwidth to retrieve and characterize the orientation and angular backscattering properties of the surface and subsurface features. MCoRDS data acquired over two distinct test areas in Greenland are used to demonstrate the specular backscattering properties of the ice surface and of the internal layers, as well as the much wider angular response of the bedrock. The coupling of internal layers' orientation with the bed topography is shown to increase with depth. Spectral filtering can be used to increase the SNR of the internal layers and for mitigating the surface multiple.

**1  Introduction**

Radio-echo sounding (RES) is a well established technique for remotely measuring the thickness of ice sheets. The use of synthetic aperture radar (SAR) focusing allows to improve gain and azimuth resolution of the echograms. Overall, state-of-the-art SAR processing offers information about the spatial properties of the ice sheet and the strength of the response, which is used to determine ice thickness, internal layers' orientation and bedrock conditions, i.e. presence or absence of water. Several SAR algorithms were offered for focusing RES data, among them 1-D matched filtering (Legarsky et al., 2001), the $\omega - k$ migration (Gogineni et al., 2001), 2-D matched filtering (Heliere et al., 2007; Peters et al., 2007), and multilook time-domain back-projection (Mishra et al., 2016).

Previous studies of angular backscattering properties of the ice sheet and bed are performed in (Smith et al., 2010; Jezek et al., 2009). Smith et al. (2010) estimate an optimal value for the SAR beamwidth based on the bedrock signal-to-noise ratio (SNR). Jezek et al. (2009) offer a technique for studying the backscattering properties of the ice sheet and bed using a special subaperture SAR approach. The authors study the dependency of the surface and bed return power on the incidence angle, the effect of the surface slope on the surface return power, they show that the response of the internal layers is specular, and propose incoherent presumming of subapertures to improve the SNR of internal layers.

In this paper we introduce a new flexible technique to analyze the angular backscattering properties of the ice-sheet and bed, which can be applied to previously conventionally SAR focused echograms. Better understanding of those properties allows us to offers novel strategies for improving internal layer and possibly bed SNR, to mitigate the surface multiple return, and

to train sparsyfing dictionaries for model-based cross-track focusing methods such as (Wu et al., 2011; Heister and Scheiber, 2016).

This paper begins with a description of the employed SAR focusing algorithm for RES data in Sect. 2. After that we introduce the technique for analyzing angular backscattering properties of the ice-sheet and bed in Sect. 3. In Sect. 4 we analyze the
5  results of processing two RES datatakes collected by the Center for Remote Control of Ice Sheets (CReSIS), Kansas, USA using their Multichannel Coherent Radar Depth Sounder (MCoRDS) during the Greenland campaign in 2008 (Gogineni, 2012). Based on the results of Sect. 4, we discuss and demonstrate approaches for improving internal layers' visibility and for mitigating the surface multiple in Sect. 5. Finally, summary and conclusions are given.

**2  SAR focusing**

10  We perform SAR focusing of RES data using a modification of the range-Doppler algorithm. The processing is done in overlapping azimuthal blocks with each block processed as described in Algorithm 1. For each block we assume the platform to fly with a constant velocity $v$, the ice surface to have a constant along-track slope $\psi$, and the ice sheet to have a constant refractive index $n_{\mathrm{ice}} = 1.78$. We also assume that the electromagnetic wave propagation obeys Snell's law for a two-layer air-ice model shown in Fig. 1. The number of azimuthal samples in each block is selected to satisfy at least twice the desired SAR
15  beamwidth of $\Delta\theta = 30°$. We additionally assume that the azimuthal antenna pattern is broad enough so that its variation for incidence angles in the interval $\theta = [-15°, 15°]$ can be safely ignored.

We now describe the inputs for Algorithm 1 using the notation where $\tau$ denotes range time, $f_\tau$ denotes range frequency, $\eta$ denotes azimuth time, and $f_\eta$ denotes azimuth frequency. The matched filter for range compression $\mathrm{H_{RC}}$ is a complex conjugate of the Fourier transform of the transmitted signal weighted by the Hann function. The motion compensation filter $\mathrm{H_{MOCO}}$ only
20  corrects for a vertical component of the platform's deviation from a linear reference track in range frequency domain.
* * *
**Algorithm 1** SAR Focusing
* * *
**Require:** raw data DATA, filters $\mathrm{H_{RC}}$, $\mathrm{H_{MOCO}}$, and $\mathrm{H_{REF}}$, amount of RCM $\Delta R_{\mathrm{RCM}}$.
**Ensure:** SAR focused echogram $\mathrm{DATA_{SAR}}$
  1: $\mathrm{DATA} := \mathrm{FFT_{range}}(\mathrm{DATA})$
  2: $\mathrm{DATA} := \mathrm{DATA} \cdot \mathrm{H_{RC}} \cdot \mathrm{H_{MOCO}}$
  3: $\mathrm{DATA} := \mathrm{IFFT_{range}}(\mathrm{DATA})$
  4: $\mathrm{DATA} := \mathrm{FFT_{azimuth}}(\mathrm{DATA})$
  5: **for** $f_\eta \in [-B_{\mathrm{az}}/2, B_{\mathrm{az}}/2]$ **do**
  6:    $\mathrm{DATA}[:, f_\eta] := \mathrm{interp}(\mathrm{DATA}[:, f_\eta], \Delta R_{\mathrm{RCM}}[:, f_\eta])$
  7: **end for**
  8: $\mathrm{DATA} := \mathrm{DATA} \cdot \mathrm{H_{REF}}$
  9: $\mathrm{DATA_{SAR}} := \mathrm{IFFT_{azimuth}}(\mathrm{DATA})$
 10: **return** $\mathrm{DATA_{SAR}}$
* * *
[Figure]

[revised manuscript text omitted]

---

## Author Comment (AC1) · 12 Jun 2018

We would like to thank the anonymous referee for the constructive comments and suggestions. We appreciate concise suggestions to improve the scientific contribution of our manuscript for The Cryosphere readers. We would also like to thank the referee for the provided references, which helped us to better understand the needs and concerns of the glaciologists working with the RES data interpretation

Hereby we provide the answer to the referee's response.

**1 Answer to General Comments**

**Referee, General Comments:** *In this manuscript, the authors present a novel radar focusing method that improves image quality based on the scattering properties of the ice surface, englacial reflectors, and ice bottom. This method can be used to improve the signal to noise ratio of englacial reflectors, can aid in the elimination of surface clutter and surface multiples, and can be used to diagnose the roughness characteristics of the surface and subsurface reflectors. The authors present the algorithm in sufficient detail to reproduce the results of their analysis, but the discussion of the **scientific** impact of the algorithm is limited – at present, this work is more appropriate for IEEE. To publish this work in The Cryosphere, the authors should include either (a) substantial discussion of the scientific utility of the new algorithm, or (b) more rigorous interpretation of their test data.*

**Authors**: We would like to stress, that the initial goal we've pursued in this paper was to provide glaciologists who use SAR RES data with a tool for analyzing angular scattering characteristics of the ice sheet and bed. The methods for improving the SNR of the internal layers and for mitigating surface multiples are provided as illustrative examples on how the knowledge of the aforementioned scattering characteristics can be utilized in practice.

The referee suggests two ways to improve the scientific of the publication. With regards to the second way ("more rigorous interpretation of their dataset"), in our opinion the restricted length and single flight direction configuration of the datasets analyzed is insufficient to provide a comprehensive glaciological interpretation of the scenes; additionally, our background is electrical engineering, and we do not posses sufficient expertise that would allow us to go a step further and provide a substantial glaciological interpretation of the datasets.

Thereby we would like to follow the first way ("substantial discussion of the scientific utility of the new algorithm") by including a discussion section on the scientific utility of the offered approach in a revised version of the manuscript (more on that in the last part of our response). By doing so, we hope that the revised version of the manuscript will contribute to the fruitful discussion between electrical engineers and the application scientists in glaciology.

**2 Answer to Specific Comments**

**Referee, Specific Comments, paragraph 1:** *The aims and scope of The Cryosphere sets standards for originality and impact of the published works. These are areas of weakness for the manuscript at present. Direction of arrival analysis is already well established in the literature (Al-Ibadi et al., 2017; Jezek et al., 2009; Wang et al., 2016). The authors do present a novel numerical scheme for inferring direction of arrival in the along-track direction. But their primary analysis is to show that englacial layers are more specular than the bed, and that deep layers are more conformal with the bed than shallow ones, two observations already discussed widely in the literature.*

**Authors**: We agree that the direction of arrival estimation (DOAE) is a well established technique in the field of RES of the terrestrial ice sheets. However, the papers cited by the referee have different objectives compared to ours and neither analyze nor directly address angular scattering characteristics of the ice sheet and bed. Al-Ibadi et al. (2017) apply a DOAE algorithm in the cross-track dimension in order to estimate the position of the bedrock scatters in the cross-track and after that generate a digital elevation model of the ice bottom. Jezek et al. (2009) offer a method that can be used to measure spatial reflectivity and 3-D surface and basal topography of the ice sheets; examples of the 3-D basal and bottom topographies are presented in the paper. Wang et al. (2016) discuss the development and performance of a new multichannel wideband radar for RES; a part of the processing chain for the radar includes use of DOAE in the cross-track dimension for surface clutter suppression and maximization of the signal's power coming from the nadir.

According to our knowledge and confirmed by the comment of the referee #2, so far a paper summarizing angular scattering properties of the ice sheet and bed is missing in the literature.

**Referee, Specific Comments, paragraph 2:** *There are few issues with the information presented, which makes this manuscript challenging to review. The authors cite (but do not pursue) two possible scientific applications*

*of their method   (1) to characterize bed roughness or subglacial hydrology (Schroeder et al., 2013) and (2) to better estimate layer slopes (Macgregor et al., 2015) and potentially relate those slopes to ice dynamics. To make this manuscript more appropriate in The Cryosphere, the authors could provide more substance in a number of ways: discuss the ice dynamic understanding derived from resolving steep englacial layers (Hindmarsh et al., 2006; Holschuh et al., 2017), spatially constrain the roughness characteristics in their test data and relate those to the underlying geology or ice flow behavior (Schroeder et al., 2014), or work toward a better understand the waveform characteristics of englacial reflectors (Drews et al., 2012) or diffuse scatterers (Jordan et al., 2017). Right now, the manuscript simply defines an algorithm and applies it  there is essentially no interpretation of results. While the summary and conclusions might be interesting to radioglaciologists who regularly process radar data, they are unlikely to be interesting to radar-data end users or a general Cryosphere audience.*

**Authors**: In order to improve the scientific value of the manuscript for The Cryosphere readers, we would like to complement our paper with a discussion section on the scientific utility of the offered approach.

For the bed roughness characterization we plan to spatially constrain the bedrock response beamwidth in our datasets and compare the results with the roughness characteristic (specularity content) derived using the method presented by Schroeder et al. (2013).

Similarly, for the layer slope estimation we plan to compare the performance of our approach with at least one of the methods proposed by MacGregor et al. (2015), focusing on challenging areas where the slope of internal layers varies strongly in azimuth.

Additionally, we would also briefly discuss the other references provided by the reviewer.

**Referee, Technical Corrections:** *See the referee's comments.*

**Authors**: We agree with the need for the technical corrections suggested by the referee and will incorporate them into a revised version of the manuscript.

**References**

M. Al-Ibadi, J. Sprick, S. Athinarapu, T. Stumpf, J. Paden, C. Leuschen, F. Rodriguez, M. Xu, D. Crandall, G. Fox, D. Burgess, M. Sharp, L. Copland, and W. V. Wychen. DEM extraction of the basal topography of the canadian archipelago ICE caps via 2d automated layer-tracker. In *2017 IEEE International Geoscience and Remote Sensing Symposium (IGARSS)*. IEEE, jul 2017. doi: 10.1109/igarss.2017.8127114.

K. Jezek, P. Gogineni, X. Wu, E. Rodriguez, F. Rodriguez, and A. Freeman. Global ice sheet mapping observatory: Airborne experiments. In *2009 IEEE Radar Conference*. IEEE, 2009. doi: 10.1109/radar.2009.4976977.

J. A. MacGregor, M. A. Fahnestock, G. A. Catania, J. D. Paden, S. P. Gogineni, S. K. Young, S. C. Rybarski, A. N. Mabrey, B. M. Wagman, and M. Morlighem. Radiostratigraphy and age structure of the greenland ice sheet. *Journal of Geophysical Research: Earth Surface*, 120(2):212–241, feb 2015. doi: 10.1002/2014jf003215.

D. M. Schroeder, D. D. Blankenship, and D. A. Young. Evidence for a water system transition beneath thwaites glacier, west antarctica. *Proceedings of the National Academy of Sciences*, 110(30):12225–12228, jul 2013. doi: 10.1073/pnas.1302828110.

Z. Wang, S. Gogineni, F. Rodriguez-Morales, J.-B. Yan, J. Paden, C. Leuschen, R. D. Hale, J. Li, C. L. Carabajal, D. Gomez-Garcia, B. Townley, R. Willer, L. Stearns, S. Child, and D. Braaten. Multichannel wideband synthetic aperture radar for ice sheet remote sensing: Development and the first deployment in antarctica. *IEEE Journal of Selected Topics in Applied Earth Observations and Remote Sensing*, 9(3):980–993, mar 2016. doi: 10.1109/jstars.2015.2403611.

---

## Author Comment (AC2) · 12 Jun 2018

Response to John Paden, Referee #2

The authors would like to thank John Paden for his valuable comments and suggestions, complying with which, we believe, will improve the quality of the paper. We are going to add descriptive information about the algorithm, add and update the references in the revised version of the paper. Hereby we provide our response to the main comments from the supplement document of your review.

**Referee, p. 5, line 20, Eq. 10:** *Explain why this is here.*

**Authors**: The comment is regarding the presence of the refractive index of ice in the Eq. 10. The equation calculates the slope of the bed and internal layers from the corresponding geometry in a SAR focused echogram. We compare slopes derived using Eq. 10 with the slopes derived from azimuthal spectrum of SAR focused echograms (Eq. 8). The latter is calculated assuming wave propagation in the air, so to compare the two, we need to adjust the former to the air case. We agree that this notation might be counterintuitive, and the alternative would be to adjust for the propagation media in Eq. 8.

**Referee, p. 11, caption of Fig. 8:** *Recommend to include a plot of internal layer SNR improvement.*

**Authors**: As we didn't correct for the antenna diagram during SAR processing, SNR improvement for internal layers will vary depending on the incidence angle for a particular layer. The estimated 10dB improvement we've declared assumes a uniform antenna pattern in azimuth. We intend to include an explanatory illustration and refined estimation for SNR improvement in the revised version of the paper.

**Referee, p. 11, line 10 and 11:** *Provide parameters for this.*
*Please clarify what filter was used? Is it a boxcar filter response (zero everywhere except inband where it is one)? That is what it sounds like from the description.*

**Authors**: We will provide parameters and an illustrative example of the piecewise linear regression used for the SNR improvement of the internal layers in the revised version of the manuscript. As for the filter, we applied a depth dependent bandpass filter in azimuth spectral domain, which will be described in more detail.

---

## Author Response (AR1)

We would like to thank the anonymous referee for the constructive comments and suggestions. We appreciate concise suggestions to improve the scientific contribution of our manuscript for The Cryosphere readers. We would also like to thank the referee for the provided references, which helped us to better understand the needs and concerns of the glaciologists working with the RES data interpretation

Hereby we provide the answer to the referee's response.

**1 Answer to General Comments**

**Referee, General Comments:** *In this manuscript, the authors present a novel radar focusing method that improves image quality based on the scattering properties of the ice surface, englacial reflectors, and ice bottom. This method can be used to improve the signal to noise ratio of englacial reflectors, can aid in the elimination of surface clutter and surface multiples, and can be used to diagnose the roughness characteristics of the surface and subsurface reflectors. The authors present the algorithm in sufficient detail to reproduce the results of their analysis, but the discussion of the **scientific** impact of the algorithm is limited – at present, this work is more appropriate for IEEE. To publish this work in The Cryosphere, the authors should include either (a) substantial discussion of the scientific utility of the new algorithm, or (b) more rigorous interpretation of their test data.*

**Authors**: In order to improve the scientific impact of the paper we have follow the referee's suggestion to include a "substantial discussion of the scientific utility of the new algorithm" in the revised version of the manuscript. We introduced a new section on "Scientific utility of large beamwidth SAR processing". There we discussed practical benefit of the large beamwidth SAR processing of RES data for two scientific applications, namely basal water detection and estimation of internal layer slope.

**2 Answer to Specific Comments**

**Referee, Specific Comments, paragraph 1:** *The aims and scope of The Cryosphere sets standards for originality and impact of the published works. These are areas of weakness for the manuscript at present. Direction of arrival analysis is already well established in the literature (Al-Ibadi et al., 2017; Jezek et al., 2009; Wang et al., 2016). The authors do present a novel numerical scheme for inferring direction of arrival in the along-track direction. But their primary analysis is to show that englacial layers are more specular than the bed, and that deep layers are more conformal with the bed than shallow ones, two observations already discussed widely in the literature.*

**Authors (our answer to this part of the comment has not changed):** We agree that the direction of arrival estimation (DOAE) is a well established technique in the field of RES of the terrestrial ice sheets. However, the papers cited by the referee have different objectives compared to ours and neither analyze nor directly address angular scattering characteristics of the ice sheet and bed. Al-Ibadi et al. (2017) apply a DOAE algorithm in the cross-track dimension in order to estimate the position of the bed scatters in the cross-track and after that generate a digital elevation model of the ice bottom. Jezek et al. (2009) offer a method that can be used to measure spatial reflectivity and 3-D surface and basal topography of the ice sheets; examples of the 3-D basal and bottom topographies are presented in the paper. Wang et al. (2016) discuss the development and performance of a new multichannel wideband radar for RES; a part of the processing chain for the radar includes use of DOAE in the cross-track dimension for surface clutter suppression and maximization of the signal's power coming from the nadir.

According to our knowledge and confirmed by the comment of the referee #2, so far a paper summarizing angular scattering properties of the ice sheet and bed is missing in the literature.

**Referee, Specific Comments, paragraph 2:** *There are few issues with the information presented, which makes this manuscript challenging to review. The authors cite (but do not pursue) two possible scientific applications of their method – (1) to characterize bed roughness or subglacial hydrology (Schroeder et al., 2013) and (2) to better estimate layer slopes (Macgregor et al., 2015) and potentially relate those slopes to ice dynamics. To make this manuscript more appropriate in The Cryosphere, the authors could provide more substance in a number of ways: discuss the ice dynamic understanding derived from resolving steep englacial layers (Hindmarsh et al., 2006; Holschuh et al., 2017), spatially constrain the roughness characteristics in their test data and relate those to the underlying geology or ice flow behavior (Schroeder et al., 2014), or work toward a better understand the waveform characteristics of englacial reflectors (Drews et al., 2012) or diffuse scatterers (Jordan et al., 2017). Right now, the manuscript simply defines an algorithm and applies it – there is essentially no interpretation of results. While*

*the summary and conclusions might be interesting to radioglaciologists who regularly process radar data, they are unlikely to be interesting to radar-data end users or a general Cryosphere audience.*

**Authors**: In order to improve the scientific value of the manuscript for The Cryosphere readers, we have complemented the revised version of the manuscript with a discussion section on the scientific utility of the offered approach. We analyzed applicability of the bed response beamwidth for the bed specularity characterization and concluded, that the beamwidth is not well suited for that task, as a bed response does not necessarily have a prominent single peak. We offered to use the variance of the bed response instead. We compared the use of specularity content (Schroeder et al. (2013)) with the variance, and concluded that the use of the variance can potentially lead to a better detection and mapping of subglacial water bodies. This is due to the fact the variance is able to detect a specular bed response which is shifted away from the origin of the incidence angle axis, whereas depending on the amount of the shift the specularity content fails to do so.

We also discussed the effect of internal layer misregistration in azimuth in the range compressed as compared to the SAR focused RES data. We concluded that the use of the SAR focused data is critical for the proper estimation of internal layer slopes, especially in case of the strongly tilted deep layers.

**Referee, Technical Corrections:** *See the referee's comments.*

**Authors**: We agreed with the need for the technical corrections suggested by the referee and have incorporated them into a revised version of the manuscript. Section 2 has been re-written to be more comprehensible to the Cryosphere readers. We provided a general explanation of pulse compression, motion compensation, and range cell migration correction procedures. Range and azimuth has been included in Fig. 1. The notation in Eq. (2) and Fig. (1) is now consistent. Section 5 has been re-written and complemented with additional figures to better explain the internal layer filtering and the surface multiple mitigation.

[revised manuscript text omitted]

---

## Author Response (AR2)

Dear Dr. MacGregor,

we would like to thank you for the valuable comments and propositions. Hereby we provide our response to your review.

**Editor**: *Title: Is "Benefits of" really necessary here, given that the processing described is uncommon at best". I think those two words could be dropped.*

**Authors**: We agree that the "Benefits of" could be dropped; we changed the paper's name accordingly.

**Editor**: *2/9: "...highest SNR were then selected from further processing, aimed at improving the SNR of weak bed echoes in outlet glaciers." Don't include the unnecessarily negative remainder of this paragraph.*

**Authors**: We dropped the last sentence of the paragraph to remove the negative reminder.

**Editor**: *2/25 Approximately how long in azimuth are each of these "blocks"? That information is needed for the reader to assess whether the surface slope can reasonably be assumed to be uniform over that distance, which has an effect on refraction through the ice.*

**Authors**: Each azimuth block was approximately $8000\,\mathrm{m}$ long (which is added to the text), with an approximately $4000\,\mathrm{m}$ overlap between the blocks.

**Editor**: *6/9: How much of the variability in apparently layer slope is due to the layer tracing itself? About 1 degree?*

**Authors**: We'd like to describe the way we calculated the internal layer and the bed slopes in the paper. First we manually track the features on an intensity SAR echogram to get the feature depth for each azimuth position $d_{\mathrm{layer/bed}}(x)$. The pixels are spaced uniformly with depth spacing $\Delta d = 3\,\mathrm{m}$ and azimuth spacing $\Delta x = 14\,\mathrm{m}$. After that we smooth the values of $d_{\mathrm{layer/bed}}(x)$ using Savitzky-Golay filter (window size is 51 samples, filter order is 3). We then numerically calculate the derivative of $d_{\mathrm{layer/bed}}(x)$ (as discrete difference), scale it by $n_{ice}/\Delta x$ and pass the result to $\tan^{-1}()$ function. The variability due to such tracing for the surveys described in the paper must be about $\pm 2$ degree.

**Editor**: *6/11: What does it mean to "rebin the echogram in azimuth by a factor of 8"? Are the data decimated of interpolated? Clarify.*

**Authors**: We reformulated this sentence. The processing meant here is incoherent averaging of SAR echograms in azimuth followed by downsampling.

**Editor**: *6/11: Here a ramp of 15 dB/km is used, whereas later on 20 dB/km is used. A bit confusing and better to stick with one value. 20 dB/km is the reference value for ice sheets although it is probably actually lower in the regions you surveyed.*

**Authors**: For the sake of consistency, we regenerated the images using ramps of $20\,\mathrm{dB}\,\mathrm{km}^{-1}$ for both surveys.

**Editor**: *14/24-30: Rephrase these sentences. RES observations of internal layers must be due to changes in electrical properties, otherwise they wouldn't reflect (unless one can somehow mount an argument that they are due to changes in magnetic properties...). Further, these layers are clearly generally isochrones, as evidenced by MacGregor et al. (2015) and many others, and the cited studies describe a debate that is now mostly closed. Finally, it is best to avoid the term "manually" as most layer-picking algorithms are semi-automatic, rather than purely manual.*

**Authors**: We rephrased the sentences to make clear, that the internal layers observed in RES data we analyze are due to changes in electrical permittivity of the ice-sheet; we also cite Hempel et al (2000) to support the fact that the internal layers are generally iscohrones.

**Editor**: *14/31-16/10: Interesting additional argument and useful illustrations. It may be worth noting that most layer slope methods are only used to predict the internal layering, which then simplifies tracing, rather relying on those layer slopes entirely.*

**Authors**: We added a sentence to clarify that most layer tracking algorithms use slope information as auxiliary input to simplify the subsequent tracing.

**Editor**: *14/8: It appears more logical to cite Schroeder et al. (2013) here also, since that study did exactly what is described in this sentence.*

**Authors**: We originally cited Schroeder et al. (2013) in this sentence. There is a reference to Schroeder et al. (2014) at the end of the paragraph as it contains values of typical height of the platform above the surface and ice thickness for the conducted surveys.

**Editor**: *In general, the figures are too often described in the text as opposed to either in the figure caption or in the figure itself (using, e.g., a legend). Shorten in-text figure descriptions as necessary.*

**Authors**: We extended captions of Fig. 5, 6, 7, 8, 10, and 11; the in-text description of the figures has been shortened accordingly. However we have left captions of Fig. 3 and 4 unchanged as their dimensions restrict addition of the caption text, and for the readers' convenience we think it's better to keep the subfigures together.

Best regards,

Anton Heister and Rolf Scheiber

**A list of the relevant changes made to the previous version of the manuscript**:

1. corrected spelling errors

2. added information about length of azimuth blocks for SAR processing (Algorithm 1)

3. added an equivalent to $n_{\mathrm{ice}}$ real part of relative permittivity $\varepsilon_{\mathrm{ice}}$ to the description of SAR processing

4. used "semi-atomatic" and "automatic" instead of "manual" while referring to the layer tracking algorithms where applicable

5. clarified the averaging and downsampling involved in generation of SAR echograms in Fig. 3(a) and Fig. 4(a)

6. we reformulated the paragraph describing internal layers tracking; two outdated references were removed, an up-to-date reference (Hempel et al. 2000) was added

7. for all figures put axis units in parentheses

8. added colorbars with units in Fig. 8 and 10, also added descriptions of the top and bottom panels; extended capture text

9. Fig. 3(a) and Fig. 4(a): used same value for the amplitude ramp of $20\,\mathrm{dB\,km^{-1}}$ for both figures; tracked internal layers of interest with solid red lines; marked azimuth positions referenced in Fig. 5 and Fig. 6 with vertical lines

10. Fig. 9 changed notation for azimuth frequency from $f_{\mathrm{az}}$ to $f_{\eta}$, as used elsewhere in the text

11. Fig. 11 added colorbars with units to subfigures (a) and (c), also added descriptions of the panels; extended capture text

[revised manuscript text omitted]

Figure 5 shows the dependency of the return power of the previously selected internal layer and bed at four fixed azimuth positions. Those particular positions are selected to demonstrate the variety of shapes of reflective signatures for the bed and the persistent signature shape for the internal layer.

The full bandwidth echogram for track 2 is shown in Fig. 4(a), where we add a depth-dependent amplitude ramp of $20\,\text{dB}\,\text{km}^{-1}$.

5 Here the bed topography varies more as compared to track 1, the internal layers are visible close to the bed with gaps appearing at azimuth positions where the absolute value of bed slope is the highest; the surface multiple is also present in the echogram.

Figure 4(b) shows the normalized reflectivity power of the surface. The surface response is similar to the one for track 1, with higher variation of $\theta_{\max(I)}$ occurring starting from azimuth $x = 65\,\text{km}$.

Reflective properties of a single internal layer tracked with a solid red line in Fig. 4(a) are shown in Fig. 4(c). Here we select

10 a shallow layer because $\theta_{\max(I)}$ for deeper layers would lie outside the interval $\theta_n \in [-14°, 14°]$ previously selected in Sect. 3. The incidence angle $\theta_{\max(I)}$ in Fig. 4(c) varies stronger and more frequently as compared to that in Fig. 3(c).

Figure 4(d) is plotted similarly to Fig. 3(d). As expected, we observe larger color gradients for internal layers for track 2, whereas incidence angles of the surface multiple lie around $\theta_n \approx 0°$ in white, corresponding to the ice surface.

The normalized power of the bed response for track 2 is shown in Fig. 4(e).

15 In Fig. 5 we compare the responses of the previously selected internal layer and bed at four fixed azimuth positions. We again observe specular reflections from the internal layer and wider reflections from the bed, with $\theta_{\max(I)}$ for the bed and the internal layer positively correlated for each selected position.

For both tracks the mean value of the surface and the layer beamwidth at the $-6\,\text{dB}$ level is $2.2°$. The shape of the bed response varies and does not necessarily have a prominent single peak, therefore we do not calculate its beamwidth. We

[Figure]

**Figure 6.** Internal layer (top panel) and bed (bottom panel) returns for track 2 at azimuth positions $x = (3, 25, 32, 54)\,\mathrm{km}$, marked with vertical red lines in Fig. 4(a). Quadratic interpolation was applied to smooth the signatures.

[revised manuscript text omitted]
, frequently observed in RES echograms, are widely attributed to the changes of electrical conductivity within the ice sheet. Owing to the fact that the internal layers are considered isochrones (Hempel et al., 2000), when tracked in RES data, they provide information about changes in the ice flow (Siegert, 2004) and snow accumulation rate in the past (Fahnestock et al., 2001). The availability of a large amount of RES data and the fact that the semi-automatic layer tracking is prohibitively expensive (Sime et al., 2011) motivate the development of automatic layer tracking algorithms.

[Figure]

(a) Normalized bed response.

(b) Specularity content and variance of the bed response.

(c) Normalized bed response.

(d) Specularity content and variance of the bed response.

**Figure 11.** Specularity content and variance as measures of specularity of the bed return for track 2. For comparison we take two parts of track 2 each 2.3 km long. The power of the angular energy returns (a) and (c) is normalized with respect to the highest value of the distribution at each azimuth position. The area inside the dashed horizontal lines in (a) and (c) corresponds to $\Delta\theta_1$.

Some of the tracking algorithms use reflection slope to predict the internal layering, which in turn simplifies subsequent tracing. MacGregor et al. (2015) introduce two new methods for the slope estimation, namely horizontal phase gradient and Doppler centroid methods. Both methods use coherent RES data (phase preserved). As stated in (MacGregor et al., 2015) currently the data have been range compressed, but without SAR focusing. The use of range compressed as opposed to SAR focused RES data might lead to erroneous estimation of the slope due to the displacement of internal layers in azimuth. The

[Figure]

**Figure 12.** Azimuth displacement of a tilted internal layer in range-compressed data.

displacement is due to the fact that, the internal layer reflection is specular, which in turn means that prior to SAR focusing the return from an internal layer appears at the azimuth position where the incident energy is normal to the layer's surface. SAR focusing registers the return at its zero Doppler position, which corresponds to the nadir direction.

Figure 12 schematically illustrates this effect with two examples. For the sake of simplicity we ignore the ray bending due to the difference in refractive properties of the air and ice. The azimuth positions at which the layer returns are registered in range compressed and SAR focused data are marked with circles and squares, respectively. A convex internal layer in Fig. 12 (a) appears stretched in azimuth in range compressed data, whereas a concave internal layer in Fig. 12 (b) appears shrunk (in extreme cases reflections from the left and the right would be overlaid).

The amount of the displacement $\Delta x$ depends on the layer's geometric slope $\alpha$, its depth $d$, platform height above the surface $R_0$, the refractive index of the ice $n_{\mathrm{ice}}$, and the surface slope $\psi$. When $\psi = 0°$, the displacement is calculated as

$$\Delta x = R_0 \tan(\sin^{-1}(n_{\mathrm{ice}} \sin \alpha)) + n_{\mathrm{ice}} d \tan \alpha. \tag{12}$$

Figure 13 shows the dependency of $\Delta x$ upon the layer geometric slope $\alpha$ and depth $d$ when the height over the surface is $R_0 = 800\,\mathrm{m}$, the ice refractive index is $n_{\mathrm{ice}} = 1.78$, and $\psi = 0°$.

[revised manuscript text omitted]

---

## Author Response (AR3)

Authors' Response

As the editor decision on the paper is to publish as is (22 Aug 2018), this manuscript version does not contain any changes.